# Contradiction Management for Medical RAG Systems – ContRAG-Med

## Problem Statement

Medical retrieval-augmented generation (RAG) systems ground answers in external evidence: clinical guidelines, systematic reviews, primary studies, drug labels, and local protocols. These sources are heterogeneous and time-dependent; they may disagree due to guideline version drift, different patient populations, eligibility criteria, endpoints, contraindications, or dosage regimens. When conflicting facts are stored and retrieved without explicit handling, the generator may blend incompatible statements or choose an unsafe option while hiding uncertainty. This motivates a principled task of detecting, typing, and managing contradictions in medical RAG knowledge stores.

## Critical Limitations of Current Methods

1. **Implicit conflict blending** — contradictory passages are retrieved and then "averaged" during generation without conflict signaling.

2. **No contradiction typing** — pipelines rarely distinguish dosage conflicts, definition/threshold mismatches, population scope differences, or temporal conflicts.

3. **Evidence-agnostic ranking** — relevance-based retrieval may prioritize low-quality or outdated evidence over stronger recommendations.

4. **Weak provenance** — the mapping *claim* $\rightarrow$ *source* is frequently lost after chunking and post-processing.

5. **Hard-to-extract atomic claims** — extracting a set of well-formed, atomic clinical claims from free text is itself a challenging NLP problem (segmenting, decontextualizing, and normalizing entities/relations).

6. **No transparent conflict management policy** — systems lack a formal, auditable rule to select or present conflicting claims.

# Proposed Solution

We propose **ContRAG-Med**, a conflict-aware module that operates on extracted medical *claims* and their metadata. The method includes four key innovations:

1. **Provenance-preserving claim representation** with explicit evidence strength and timestamps.

2. **Weighted contradiction detection** that outputs a continuous contradiction degree.

3. **Probabilistic contradiction classification** into interpretable types to enable different handling strategies.

4. **Graph-based conflict management** with an evidence–recency priority rule and a calibrated "show-both" option for near-ties.

# Formal Setup

Let $D = \{d_1, \ldots, d_N\}$ be a set of medical documents and $C = \bigcup_{i=1}^{N} C(d_i)$ be the set of extracted claims, $|C| = n$. Each claim is a tuple

$$c = (t, \varphi, m, \mathrm{src}), \qquad m = (e, \tau),$$

where $t$ is the claim text, $\varphi$ is its semantic representation, and $m$ contains (i) an evidence level $e \in \{1, \ldots, 5\}$ (or strength of recommendation) and (ii) a timestamp $\tau \in \mathbb{R}_+$. The source field $\mathrm{src} \in D$ denotes a *document identifier* that also links to document-level metadata (e.g., publisher, source type, venue, and version), rather than the raw document text alone.

**Detection.** A scoring function $f_{\mathrm{detect}} : C \times C \to [0, 1]$ satisfies antireflexivity and symmetry, and may incorporate evidence disparity $|e(c_i) - e(c_j)|$. Define the matrix

$W \in [0,1]^{n \times n}$ by

$$W_{ij} = f_{\text{detect}}(c_i, c_j),$$

and the contradiction relation

$$(c_i, c_j) \in R_{\text{cont}} \iff W_{ij} \geq \theta.$$

**Typing.** For contradictory pairs, a classifier $f_{\text{classify}} : R_{\text{cont}} \to \Delta^{K-1}$ outputs a distribution over $K$ contradiction types. Define

$$\text{strength}(c_i, c_j) = W_{ij} \cdot \max_k \Pr(k \mid c_i, c_j),$$

and construct the weighted contradiction graph

$$G_{\text{cont}} = (V, E, w, \tau),$$

where $V = C$, $E = \{(c_i, c_j) \mid (c_i, c_j) \in R_{\text{cont}}\}$, $w(c_i, c_j) = \text{strength}(c_i, c_j)$, and $\tau$ assigns time to each claim.

**Resolution.** We use an auditable priority score that combines evidence and recency:

$$\text{priority}(c) = \alpha \frac{6 - e(c)}{5} + \beta \exp\!\left(-\gamma \left|\tau_{\text{now}} - \tau(c)\right|\right), \qquad \alpha + \beta = 1,$$

where $\alpha, \beta, \gamma > 0$. To avoid overly rigid decisions, we introduce a near-tie threshold $\delta > 0$. For a contradictory pair $(c_i, c_j) \in R_{\text{cont}}$:

if $\left|\text{priority}(c_i) - \text{priority}(c_j)\right| < \delta$, then surface both claims with a "Conflicting Evidence" flag (and p

otherwise, the higher-priority claim is used as the default, while still retaining the alternative claim for transparency.

# Practical Applications

- Auditing and cleaning medical knowledge stores by locating conflict hotspots and their causes
- Conflict-aware retrieval and reranking that downweights outdated or low-evidence contradictions
- Monitoring guideline drift and inconsistencies between guideline versions and

local protocols

- Safer answer generation with explicit provenance and calibrated presentation of multiple positions

## Conclusion

ContRAG-Med provides a formal, auditable foundation for contradiction detection, classification, and conflict management in medical RAG pipelines, enabling safer retrieval and generation under heterogeneous and evolving clinical evidence.

