# OpenReview forum: "Решение задачи обнаружения и классификации противоречий в медицинских системах с RAG — ContRAG-Med"
_mathai.club/MathAI/2026/Conference — MathAI 2026 Conference Submission_

### Official Review · Reviewer_8aeR · 2026-03-10
**Weak Accept. Promising formal framework for trustworthy medical RAG, but Abstract is in Russian, no references, short text size.**

**Rating:** 6
**Confidence:** 4

**Review:**

This short work presents general problems of retrieval-augmented generation (RAG) systems in medicine.
MATHEMATICAL RIGOR - good.
Paper defines formal setup with sets of parameters, scoring function, contradiction matrix W, graph and priority, but lacks theorems or full proofs.
NOVELTY & CONTRIBUTION - good.
Introduces ContRAG-Med with innovations: provenance-preserving claims, weighted detection, probabilistic typing, graph-based resolution with near-tie handling; original method beyond surveys
RELEVANCE TO MathAI conference - good.
Fits B=ML Theory & Methods and C=Trustworthy & Explainable AI via mathematical formalization of RAG contradictions in medical AI; strong AI-math intersection but applied focus
TECHNICAL QUALITY - good.
Sound methodology with auditable rules; formal definitions consistent, no obvious errors in extracted math; implementation details (e.g., f_"classify" ) assumed correct absent experiments.
CLARITY & PRESENTATION - good.
Structured (problem, limitations, solution, formal setup); it has repetitive titles, and no full experiments. No references at the end of the text.
AI-GENERATION RISK - high.
Generic structure (problem-limits-proposal-formal-conclusion), math breadth without deep proofs/experiments, lacks author expertise cues; plausible but suspiciously polished for short paper.

Overall Recommendation- Weak Accept.
Promising formal framework for trustworthy medical RAG, but needs proofs, experiments, and author details for stronger accept.
The text lacks references.
Original Abstract was in Russian, not properly translated.

---

### Official Review · Reviewer_vsSY · 2026-03-11
**A conceptually strong but poorly formatted draft on formal contradiction management for medical RAG systems.**

**Rating:** 3
**Confidence:** 4

**Review:**

The paper addresses an important topic — managing contradictions in medical RAG systems. The problem is clearly formulated, and the proposed solution is built upon a rigorous mathematical model. However, the work reads more like an extended abstract than a full conference paper. The main drawbacks are critical formatting violations (the template is not followed, the title and abstract in the submission are in Russian) and the absence of a reference list, which is unacceptable for a scientific paper.

Pros:
1.Relevance: The issue of hallucinations and contradictions in RAG systems for the sensitive medical domain is a key challenge in the field of Trustworthy AI.
2.Novelty of the Approach: The introduction of contradiction typing (via f_classify) and a "near-tie" mechanism (threshold δ) for surfacing both claims is noteworthy and practically interesting. This approach moves beyond a rigid selection of a single source, offering a more nuanced strategy.

Cons and Remarks:
1.Critical Formatting Violations: The paper is not prepared in LaTeX and does not follow the provided template! The title and abstract in the submission are written in Russian.
2.Missing Reference List: There are no references at the end of the paper. Citations mentioned in the text are not included in a References section.
3.Incompleteness and Lack of Validation: The work presents a purely theoretical (formal) model. It lacks any experiments, worked examples, computational complexity analysis, or comparison with baseline approaches. It is unclear how the proposed functions (f_detect, f_classify) would be implemented in practice.
4.Incomplete Description of Key Components: The paper does not clarify the semantic representation $\varphi$, how features for the f_classify classifier are constructed, or how atomic claims C are intended to be extracted — despite the authors themselves rightly noting the difficulty of this task. This lack of specificity hinders an assessment of the approach's feasibility.

---

### Official Review · Reviewer_CZrN · 2026-03-11
**A Formally Grounded but Practically Unrealized Approach to Contradiction Management in Medical RAG Systems**

**Rating:** 6
**Confidence:** 4

**Review:**

This work addresses a critically important and practical problem. As medical RAG systems move toward clinical deployment, the safe management of contradictory evidence is paramount. The paper offers a clear, formal blueprint that could influence the design of future systems and serve as a foundation for auditable and transparent AI in healthcare.

The paper introduces ContRAG-Med, a formal framework for detecting, classifying, and resolving contradictions in medical Retrieval-Augmented Generation (RAG) systems. The authors identify critical limitations in existing approaches, such as implicit conflict blending and loss of provenance. They propose a solution—a modular pipeline operating on extracted medical claims. Key features include: 1) provenance-preserving claim representation with evidence strength and timestamps; 2) a weighted contradiction detection function; 3) probabilistic classification of contradiction types; 4) a graph-based resolution module that uses a priority score (evidence level + recency) to select the best claim or present both in case of a "near-tie". The work is formally defined mathematically and aims to provide an auditable foundation for safer medical RAG applications.

The formalization of the problem is a clear strength of the work. Defining claims as tuples and contradictions via a scoring function and a graph provides a precise, unambiguous language for describing the problem. The priority function combines two key dimensions—evidence strength and timeliness—into a single tunable metric. The introduction of a "near-tie" threshold is a thoughtful addition that avoids overly rigid decisions.

The specific application of contradiction handling in RAG to the medical domain, with integrated evidence levels and temporal decay, is novel and well-motivated. The "show-both" policy for near-ties, governed by a calibrated threshold, is a creative and responsible approach for medical applications. The idea of using a contradiction graph for management, rather than just detection, is a step forward compared to simple pairwise comparison.

However, at this stage, the work remains purely conceptual. It lacks experimental validation, which makes it impossible to assess the effectiveness or reliability of the proposed methods. Key components, especially the claim extraction stage and the contradiction type classifier, are described only at a high level. There is no discussion of how to operationalize these complex NLP tasks, what models could be used, or how to evaluate them.

At present, the significance is more potential than demonstrated. The real impact will depend on successful implementation and evaluation on real-world or benchmark medical question-answering tasks.

---

### Decision · Program_Chairs · 2026-03-20

**Decision:**

Reject

**Comment:**

After careful evaluation by the Program Committee, we regret to inform you that your submission has not been accepted for presentation at MathAI 2026.

All submissions underwent a rigorous two-stage review process. Unfortunately, the reviewers identified one or more of the following concerns with your paper:

- Insufficient mathematical rigor or novelty relative to the existing body of work in the field;
- Presentation of results that substantially overlap with or rephrase previously published findings without clear original contribution;
- Significant issues with technical quality, including but not limited to broken or non-existent references, unsupported claims, or methodological gaps;
- Indications that the manuscript may have been generated with the assistance of large language models without substantial original intellectual contribution by the authors.

We received a large number of submissions this year, and the selection process was highly competitive. We encourage you to carefully consider the reviewers’ feedback (available through OpenReview), revise your work accordingly, and consider submitting an improved version to a future edition of MathAI or to another appropriate venue.

We appreciate your interest in MathAI and hope you will continue to engage with the conference community.

With kind regards,

MathAI 2026 Program Committee
International Conference on Mathematics of Artificial Intelligence
https://mathai.club
OpenReview: https://openreview.net/group?id=mathai.club/MathAI/2026/Conference
MathAI Telegram: https://t.me/MathAI_club
IAIC International AI Committee: https://t.me/iaic_world
Email: mathai.club@yandex.ru